# Operon mRNAs are organized into ORF-centric structures that predict translation efficiency

David H Burkhardt[1,2,3†], Silvi Rouskin[3,4,5†], Yan Zhang[2,6†], Gene-Wei Li[3,4,5*‡], Jonathan S Weissman[3,4,5*], Carol A Gross[2,3,6*]

[1]Graduate Group in Biophysics, University of California, San Francisco, San Francisco, United States; [2]Department of Microbiology and Immunology, University of California, San Francisco, San Francisco, United States; [3]California Institute of Quantitative Biology, University of California, San Francisco, San Francisco, United States; [4]Department of Cellular and Molecular Pharmacology, Howard Hughes Medical Institute, University of California, San Francisco, San Francisco, United States; [5]Center for RNA Systems Biology, University of California, San Francisco, San Francisco, United States; [6]Department of Cell and Tissue Biology, University of California, San Francisco, San Francisco, United States

**\*For correspondence:** gwli@mit.edu (G-WL); Jonathan.Weissman@ucsf.edu (JSW); cgrossucsf@gmail.com (CAG)

[†]These authors contributed equally to this work

**Present address:** [‡]Department of Biology, The Massachusetts Institute of Technology, Cambridge, United States

**Competing interests:** The authors declare that no competing interests exist.

**Abstract** Bacterial mRNAs are organized into operons consisting of discrete open reading frames (ORFs) in a single polycistronic mRNA. Individual ORFs on the mRNA are differentially translated, with rates varying as much as 100-fold. The signals controlling differential translation are poorly understood. Our genome-wide mRNA secondary structure analysis indicated that operonic mRNAs are comprised of ORF-wide units of secondary structure that vary across ORF boundaries such that adjacent ORFs on the same mRNA molecule are structurally distinct. ORF translation rate is strongly correlated with its mRNA structure in vivo, and correlation persists, albeit in a reduced form, with its structure when translation is inhibited and with that of in vitro refolded mRNA. These data suggest that intrinsic ORF mRNA structure encodes a rough blueprint for translation efficiency. This structure is then amplified by translation, in a self-reinforcing loop, to provide the structure that ultimately specifies the translation of each ORF.

## Introduction

Protein synthesis is the most energetically costly process in bacteria, consuming up to 50% of cellular energy. Thus, to optimize cellular efficiency, the rate of synthesis of each protein must be carefully controlled. In bacteria, operons are central to this process. Open reading frames (ORFs) with related functions are organized into operons that are transcribed as a single mRNA ensuring that operonic genes are transcriptionally co-regulated in response to various conditions (*Jacob and Monod, 1961*). Additionally, the translation of each ORF in the operon is precisely tuned to cellular need. Indeed, the rate of protein production (i.e. the translation efficiency) of adjacent ORFs on a single mRNA can vary by as much as 100-fold, and members of protein complexes encoded on a single mRNA are generally translated in proportion to their stoichiometry (*Li et al., 2014*). Understanding how the cell achieves optimal energy utilization critically depends on understanding how mRNA sequence features reliably drive ORF-specific translation.

A number of mRNA features have been identified as contributing to the rate of translation of an ORF. Both the strength and accessibility of a Shine-Dalgarno (SD) sequence upstream from the initiation codon (*Steitz and Jakes, 1975*) have been implicated in translatability. In support of the

**eLife digest** Proteins make up much of the biological machinery inside cells and perform the essential tasks needed to keep each cell alive. Cells contain thousands of different proteins and the instructions needed to build each protein are encoded in genes. However, these instructions cannot be used directly to manufacture the proteins. Instead, a messenger molecule called mRNA is needed to carry the information stored within genes to the parts of the cell where proteins are made.

In bacteria, one mRNA molecule can include information from several genes. This group of genes is called an operon and produces a set of proteins that perform a shared task. Although these proteins work together, some of them are needed in greater numbers than others. Because they are all made using information from the same mRNA, some instructions on the mRNA must be read more times than others. It is unclear how bacterial cells control how many proteins are produced from each part of one mRNA but it is thought to relate to the three-dimensional shape of the molecule itself.

Burkhardt, Rouskin, Zhang et al. have now examined the production of proteins from mRNAs in the commonly studied bacterium, *Escherichia coli*. The results showed that each set of instructions on the mRNA formed a three-dimensional structure that corresponds to the amount of protein produced from that portion of the mRNA. When this three-dimensional structure is more stable or rigid, the corresponding instructions tended to produce fewer proteins than if the structure was relatively simple and unstable.

Further investigation showed that these three-dimensional mRNA structures could form spontaneously outside of cells, suggesting that molecules other than the mRNA itself have a relatively small role in controlling the number of proteins produced. This also suggests that the entire structure of each mRNA is important and is likely to be essential for cell survival. The next step is to understand why bacteria organise their genes in this way and how the different mRNA structures control how proteins are produced. Moreover, because many bacteria are used like biological factories to produce a variety of commercially useful molecules, these new insights have the potential to enhance a number of manufacturing processes.

importance of SD accessibility, highly stable mRNA structures in direct proximity to the initiation codon diminish translatability (*de Smit and van Duin, 1990*; *Hall et al., 1982*; *Lodish, 1970*) and rare codons that disfavor RNA structure are enriched in positions immediately following the translation start site (*Bentele et al., 2013*; *Eyre-Walker and Bulmer, 1993*; *Scharff et al., 2011*). Moreover, several studies examining either synthetic ORFs with a few bases difference (e.g. alterations to GFP), or fluorescent reporter assays studying the effect of multiple codon changes in the 5' UTR and N-terminal coding sequences, find that models based on their predicted RNA structure at the translation start site are relatively successful at predicting their differences in translatability (*Goodman et al., 2013*; *Kudla et al., 2009*). Most recently, codon usage has emerged as an important variable for translation. A large study examining thousands of foreign ORFs concluded that except for the very initial nucleotides of the ORF, codon usage rather than mRNA folding propensity was the critical determinant for translatability (*Boël et al., 2016*).

While these mRNA features are of clear value for predicting the translatability of exogenously expressed ORFs, several considerations suggest that they may not capture the key features that have evolved to set the translation efficiency of endogenous genes. First, all the high-throughput studies overexpressed the mRNAs they studied, which is known to perturb the charged tRNA pool and introduce biases in codon usage (*Dittmar et al., 2005*; *Elf et al., 2003*). Second, in the Boel et al. manuscript and in other studies, mRNAs were transcribed by T7 RNA polymerase, which not only elongates significantly faster than *E. coli* RNA polymerase but also removes the influence of the many endogenous *E. coli* RNA polymerase binding proteins that modulate its elongation rate. Where examined, such RNAs exhibit altered folding patterns (*Lewicki et al., 1993*; *Chao et al., 1995*; *Pan et al., 1999*). Thus, these transcripts likely have non-native structure. Third, these studies all used foreign mRNAs, which had not been subjected to evolution for precise tuning in *E. coli*.

Finally, these studies primarily measured protein abundance, a quantity that is dependent on mRNA and protein stability as well as on the efficiency at which each ORF is translated.

The goal of this work is to understand how *E. coli* establishes the relative expression of adjacent ORFs on the same mRNA. To accomplish this, we systematically assessed the translational efficiency (TE) of every ORF mRNA and then examined which of its features (e.g. secondary structure, codon usage, and the strength of ribosome binding site) correlated with its translatability. The translation efficiency of endogenous messages in *E. coli* could be probed with existing global technologies (*Li et al., 2014*; *Oh et al., 2011*; *Ingolia et al., 2009*) and the effects of codon usage with two metrics, tAI (tRNA adaptation index) (*Tuller et al., 2010*; *dos Reis et al., 2004*) and codon influence (*Boël et al., 2016*). However, in vivo mRNA structure has not previously been empirically evaluated at the global level in *E. coli*. We therefore adapted the dimethyl sulfate (DMS)-seq technique (*Rouskin et al., 2014*), which uses next-generation sequencing to determine chemical accessibility of RNA to DMS, to *E. coli*. Our studies point to a critical role of intrinsic ORF-wide differences in mRNA structure in allowing differential translation of ORFs sharing the same operonic mRNAs.

## Results

### Development of global RNA secondary structure determination in *E. coli*

New genomic technologies enable the determination of RNA structure on a global scale (*Ding et al., 2014*; *Rouskin et al., 2014*; *Wan et al., 2014*). DMS-seq uses next-generation sequencing to determine chemical accessibility of RNA to DMS (dimethyl sulfate), a reagent that reacts with unpaired adenosine and cytosine nucleotides (*Inoue and Cech, 1985*). We adapted DMS-seq to *E. coli* to monitor global in vivo RNA structure (*Figure 1A*). By exploring the effect of coverage on reproducibility, we find that a read coverage of ~15 reads/nucleotide is sufficient for reproducible structure determination (*Figure 1B*), and used this cutoff in all subsequent analyses. Structures determined from *E. coli*-adapted DMS-seq are in excellent agreement with both the 16S rRNA crystal structure (*Figure 1C*) (*Zhang et al., 2009*), and a mutationally verified *E. coli* mRNA structure (*Figure 1D*) (*Wikström et al., 1992*).

We quantified the degree of secondary structure for each ORF using the Gini index metric, which measures the variability in reactivity of A and C residues to DMS in the region being examined (*Rouskin et al., 2014*). A low Gini index indicates a relatively even distribution of DMS-seq reads and occurs in unstructured regions of the mRNA. A high Gini index occurs when a subset of residues is strongly protected from DMS reactivity and indicates a high degree of structure (*Figure 1E*). We found that the degree of RNA secondary structure varied greatly between ORFs: a small number are nearly as structured as rRNA, whereas some are close to the denatured state (*Figure 1F*).

### *E. coli* mRNAs have intrinsic ORF-wide secondary structures

In contrast to the large variation in the degree of secondary structure among ORFs (*Figure 1F*), individual ORFs generally have fairly consistent Gini scores across their bodies – for example, the first and second halves of each ORF exhibit highly correlated Gini scores (*Figure 2A*). We tested whether the Gini scores across ORFs remained correlated in the absence of translation in two different conditions. First, we examined Gini scores of in vivo mRNA when translation initiation was inhibited with kasugamycin. We achieved rapid inhibition by using a Δ*gcvB* mutant, which has enhanced kasugamycin uptake rates (*Figure 2—figure supplement 1A*; *Shiver et al., 2016*). Using Δ*gcvB* mutant is critical for this experiment because kasugamycin uptake by wild-type (WT) cells is slow enough to allow massive degradation of mRNA before ribosomes are cleared (see extended methods for protocol and *Figure 2—figure supplement 1B–C* for method validation, including demonstrating that Δ*gcvB* does not alter global mRNA structure). Second, we examined Gini scores of purified mRNA refolded in vitro at 37°C. In both cases, the translation-independent mRNA structures obtained from DMS-seq indicated that Gini scores across the ORF mRNAs remain correlated (*Figure 2B–C*). This correlation also holds true for computationally predicted mRNA structure of ORFs (*Figure 2D*). Moreover, the degree/extent of mRNA structures (henceforth referred to as structure) determined in these various ways are highly correlated with each other (*Figure 2E–F*). We conclude that mRNA is organized in ORF-wide structures that depend on the intrinsic sequence of the mRNA.

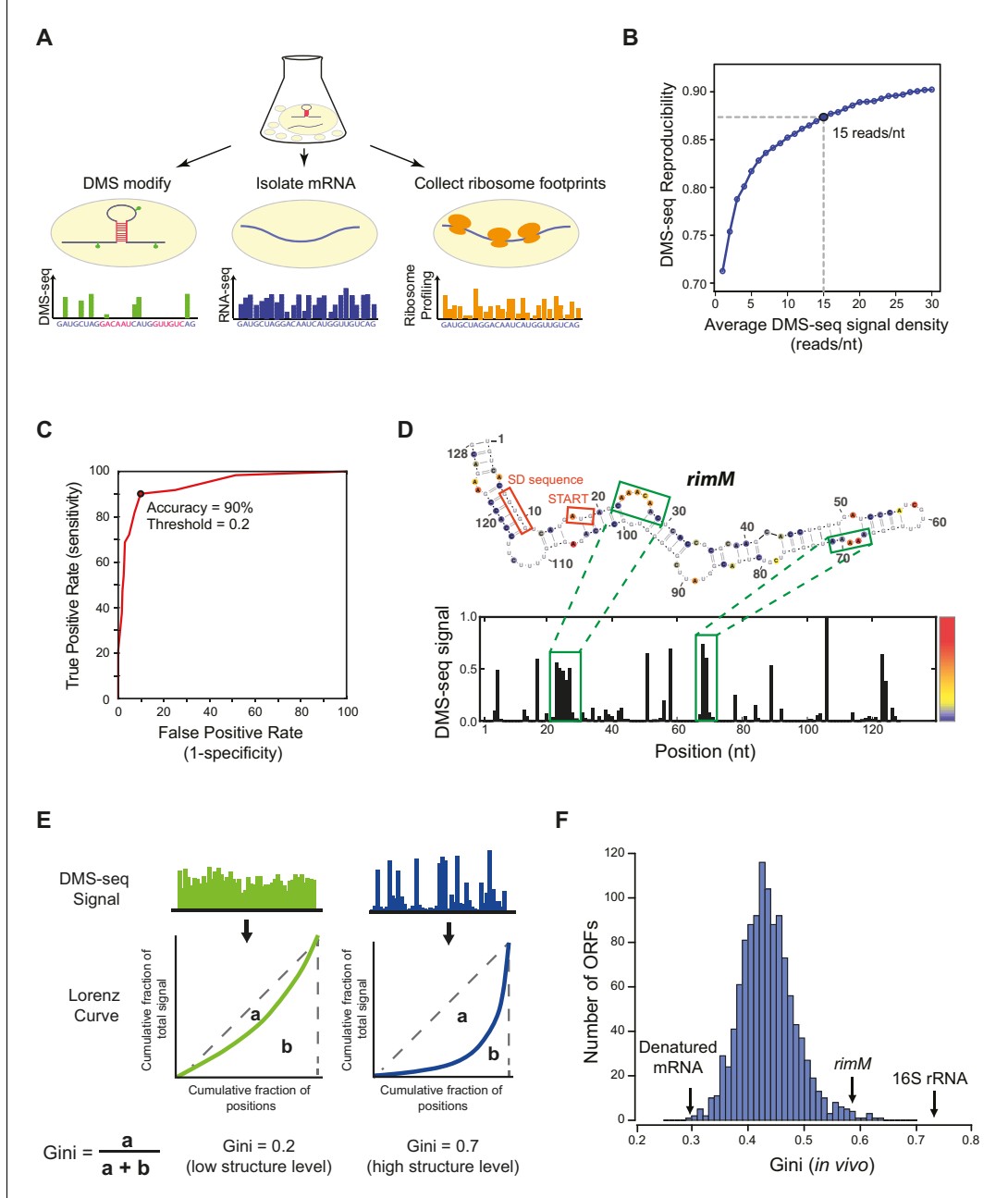

**Figure 1.** DMS-seq effectively probes RNA structures in *E. coli*. (**A**) Schematic for obtaining mRNA structure and translation efficiency using DMS-seq, mRNA-seq, and ribosome profiling from the same sample. (**B**) Plot showing the effect of DMS-seq read coverage on the reproducibility of structure determination. X-axis: DMS-seq read depth cutoff (reads/nucleotide); Y-axis: median of Pearson's R values calculated by comparing two replicates of in vivo DMS-seq signals of the first 200nt of ORFs passing the DMS-seq depth cutoff indicated in X-axis. A read coverage of ~15 reads/nucleotide is sufficient for reproducible structure determination. (**C**) Receiver operating characteristic (ROC) curve on the in vivo DMS-seq signals for A and C bases in the 16S rRNA using the *E.coli* ribosome crystal structure (*Zhang et al., 2009*) as a model. True positives are defined as bases that are both unpaired and solvent-accessible, and true negatives are bases that are paired. The total number of evaluated A/C bases is 438. Signal threshold of 0.2 has 90% agreement with the crystal structure. (**D**) Structural prediction for *rimM*. The predicted *rimM* structure is based on a minimum free-energy prediction constrained by our DMS-seq measurements, using the same 0.2 threshold used for the 16S rRNA in (**B**), which agrees with the *rimM* structure proposed and mutationally verified in *Wikström et al. (1992)*. The DMS-seq signal across *rimM* is shown below the structure. The color bar indicates the intensity of the DMS-seq signal at each position. (**E**) Calculation of the Gini index from the DMS-seq signal is indicated schematically by comparing highly structured regions to less structured regions. For a region of mRNA, the cumulative fraction of the total DMS-seq signal is plotted against the cumulative fraction of the total number of positions as a Lorenz Curve. The extent to which the curve sags below the diagonal indicates the degree of inequality of distribution, which is quantified by the Gini index defined as the ratio of the area between the diagonal line and the Lorenz Curve (a) to

*Figure 1 continued on next page*

*Figure 1 continued*

the area below the diagonal line (a + b). A high Gini index indicates high level of mRNA structure, and vice versa. (F) Histogram of Gini indices of *E. coli* ORFs calculated from in vivo DMS-seq data at 37℃. All ORFs selected have ≥15 DMS-seq reads/nt (N = 1116). The Gini index of 16S rRNA and *rimM*, and the mean of Gini indices of in vitro heat-denatured mRNAs at 95℃ are indicated.

We next examined whether structural correlation extends to adjacent ORFs on the same polycistronic (operonic) mRNA. We considered only those operons in which each ORF has an approximately equivalent mRNA levels, thus excluding those with significant internal promoters or terminators (see Materials and methods). Within operonic (polycistronic) mRNAs, the mRNA structure of adjacent ORFs can differ significantly (*Figure 2G* and *Figure 2—figure supplement 1D*), even when the start and stop codons of the adjacent ORFs overlap (*Figure 2—figure supplement 1E*). Thus, characteristic mRNA structures are a property of individual ORF mRNAs rather than of the entire polycistronic transcript.

## Translation efficiency is highly correlated with ORF mRNA structure

We next explored the relationship between the level of ORF-wide mRNA structure identified above with the TE of that ORF. We previously demonstrated that the overall rate of protein production can be accurately measured by an ORF's average ribosome footprint density (number of footprints per unit length of the ORF), showing that protein copy number per cell determined from average ribosome footprint density was in superb agreement with that obtained by individually quantifying stable proteins in *E. coli* (*Li et al., 2014*). Here, we build on that validated parameter, defining TE as the rate of protein production per mRNA, measured by normalizing average ribosome footprint density of an ORF with its mRNA abundance (i.e. RPKM of mRNA sequencing) (*Ingolia et al., 2009*; *Li et al., 2014*; *Oh et al., 2011*), with both measurements obtained from the same biological samples (see Materials and methods). Importantly, this metric is not affected by differences in either mRNA or protein abundance or stability (*Li, 2015*).

We found that the TE's of *E. coli* endogenous ORFs in operonic mRNAs were highly negatively correlated with their level of ORF-wide mRNA structure ($\rho = -0.75$, *Figure 3A*: well-translated ORFs have less mRNA structure, while poorly translated ORFs have more structure). Consistent with the fact that the ORF-wide mRNA structures of adjacent ORFs in an operon can differ significantly (*Figure 2G*), the TE's of adjacent ORFs can also differ significantly (*Figure 3B*, *Figure 3—figure supplement 1A*). Notably, ORF pairs with overlapping start and stop codons, believed to be translationally coupled (*Aksoy et al., 1984*; *Oppenheim and Yanofsky, 1980*; *Schümperli et al., 1982*; *Yates and Nomura, 1981*), show essentially as much variability in their relative translation as non-overlapping ORF pairs ($p=0.06$, K-S test, *Figure 3B*), suggesting that the extent of coupling is variable. We then expanded this analysis beyond operons to all ORFs and found that the level of mRNA structure and TE are highly anti-correlated on all endogenous open reading frames ($\rho = -0.76$, *Figure 3C*). Importantly, the Gini scores of ORFs calculated from control RNA samples without DMS modification were not correlated to TE ($\rho = 0.05$, *Figure 3—figure supplement 1B*), indicating that Gini scores calculated from DMS-seq indeed reflect the level of mRNA structure and the potential sequencing bias/noise does not contribute to the correlation between TE and mRNA structure.

Translation itself can influence mRNA structure as the helicase activity of translating ribosomes is likely to decrease the mRNA structure of highly translated ORFs more than that of poorly translated ORFs. We asked whether TE is correlated solely to the mRNA structure that results from ribosome unwinding or whether it is also correlated to the intrinsic mRNA structure that exists in the absence of translation. We find that when translation is inhibited in vivo (e.g. following kasugamycin treatment), the absolute correlation of TE to structure remains high but decreases somewhat ($\rho = -0.58$, *Figure 3D*), and that there is a small further decrease in correlation when mRNAs are refolded in vitro ($\rho = -0.48$, *Figure 3E*). Additionally, computationally predicted structures of entire ORFs also show robust correlation to their TE's ($\rho = -0.52$, *Figure 3F*). The results are very similar when we confine ourselves to the 421 ORFs with ≥15 DMS reads/nucleotide in all conditions (*Figure 4—figure supplement 2*).

We further dissected the influence of translation on ORF mRNA structure by determining how the difference in Gini score of in vivo mRNA with and without translation is related to its TE. We found

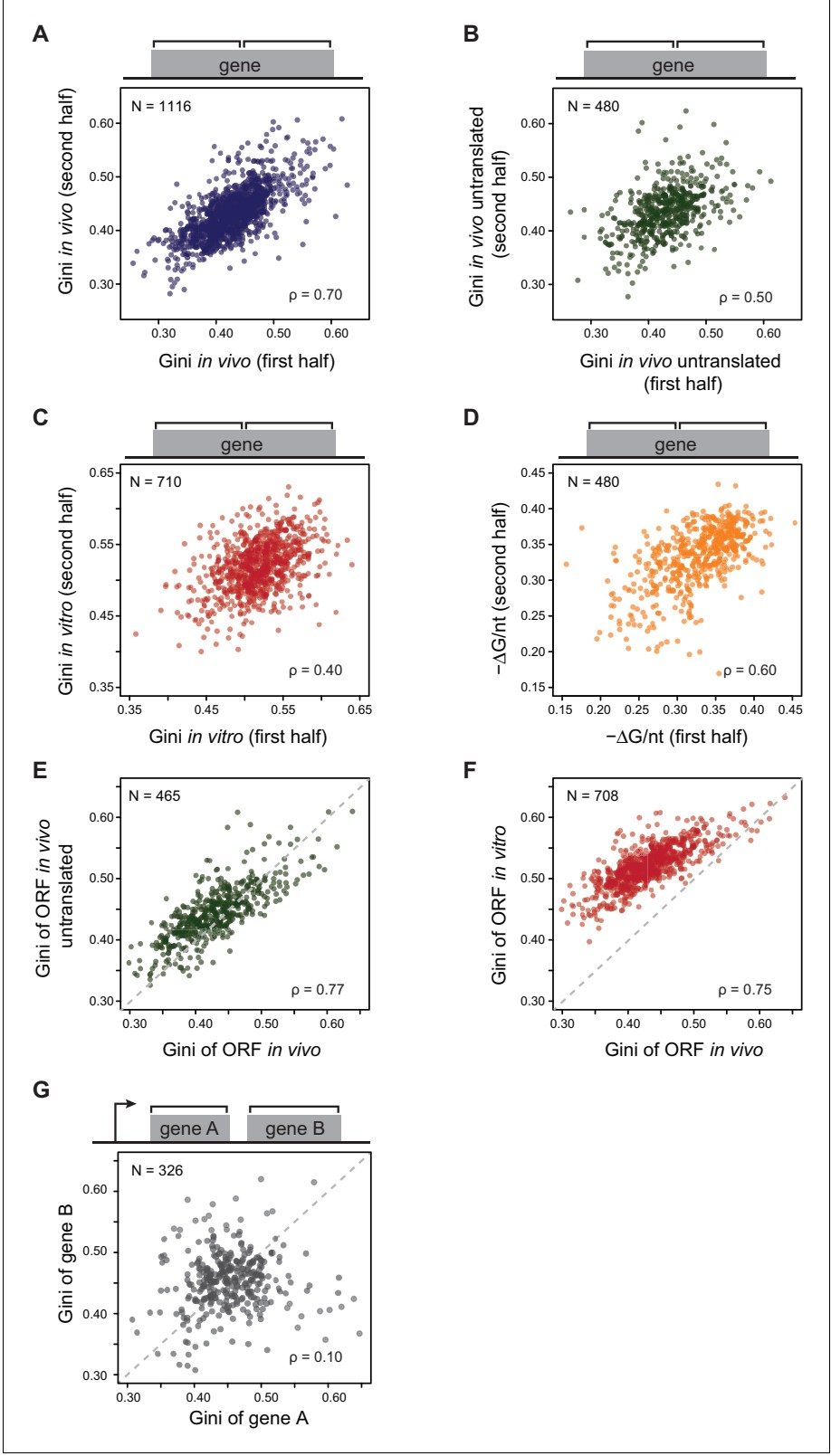

**Figure 2.** *E. coli* mRNAs have intrinsic ORF-wide secondary structures. (**A–C**) Plots comparing the Gini indices of the first half of the ORF against those of the second half of the ORF for: A. in vivo modified mRNA from cells growing at 37°C; B. in vivo modified mRNA from cells treated with kasugamycin (ksg) at 37°C (no translating ribosomes); C. in vitro mRNA modified at 37°C. In this and all subsequent figures, analysis is performed only on

*Figure 2 continued on next page*

*Figure 2 continued*

those ORFs with ≥15 DMS-seq reads per nucleotide, with N (the number of ORFs analyzed in each condition), and ρ (the Spearman's rank correlation coefficient) indicated. The ksg-treated sample has fewer ORFs passing the ≥15 DMS-seq reads/nt filter, likely due to mRNA degradation when translation is eliminated. Data calculated using different sets of ORFs are summarized in *Supplementary file 1–3*. (D) Plot comparing the computationally predicted mRNA structure (- minimum free energy / nucleotide or -ΔG/nt) of the first half of the ORF against that of the second half of the ORF for the 480 ORFs in the ksg-treated DMS-seq dataset. (E) Correlation between Gini indices of the entire ORF calculated from in vivo mRNA vs in vivo untranslated mRNA (ksg-treated cells) for the 465 ORFs in both datasets. The dashed grey line represents the y = x diagonal line. (F) Correlation between Gini indices of the entire ORF calculated from in vivo mRNA vs in vitro refolded mRNA for the 708 ORFs shared in both datasets. (G) Plot comparing Gini indices for adjacent ORFs in operons (N = 326; see Materials and methods for details). The dashed grey line represents the y = x diagonal line.

The following figure supplement is available for figure 2:

**Figure supplement 1.** mRNA structure is organized around open reading frames.

---

that there is a tendency for mRNA to be more structured (higher Gini index) in the absence of translation (*Figure 2E*) and that mRNAs with the highest TEs had the greatest difference in their Gini's (ρ = 0.52, *Figure 3G*). These data are consistent with the idea that unwinding by ribosomes contributes to the in vivo structure of highly translated genes. The decreased correlation of untranslated ORF mRNA structure to TE may result from removing the contribution of unwinding by translating ribosomes.

Previous work on ORF translatability has pointed to the important role of sequences around the ORF start site. Using the ORFs that are common in all datasets and separated by ≥20 nt from the upstream ORF, we examined the correlation of TE with the level of mRNA structure only around the start site or extending further into the ORF (*Figure 3—figure supplement 1C*). We find increasing correlation with TE as successively larger regions of the ORF are considered in the structural analysis (−20nt to 40nt, 0 to 60nt, and 0 to 100nt relative to the gene start). Notably, the correlation of TE with extent of structure in either the first or second halves of the ORF are very similar, and the highest correlation is with the Gini of the ORF-wide mRNA structure.

*In toto*, these analyses indicate that the linear sequences of bacterial mRNAs encode not only ORFs, but also ORF-wide secondary structures. These structures provide a rough blueprint for the TE of that ORF. Instructions from this blueprint are augmented by ribosomes and additional factors (see Discussion).

## Translation efficiency is less correlated with other mRNA features

We next examined the ability of the Shine-Dalgarno sequence and codon usage to predict TE. Data for all ORFs are presented in *Figure 4*, and that for the 421 ORFs in common between conditions are presented in *Figure 4—figure supplement 2*.

Consistent with earlier studies (*Li et al., 2014*), we found that the strength of the Shine-Dalgarno sequence does not have predictive power for TE, even after controlling for structure as measured by Gini index (*Figure 4—figure supplement 1A*).

Codon usage, quantified by tAI (tRNA adaptation index) (*dos Reis et al., 2004*; *Tuller et al., 2010*) modestly correlates with TE (ρ = 0.34, *Figure 4A*). Interestingly, codon usage correlates more strongly with the overall rate of translation (i.e. average ribosome footprint density, ρ = 0.61) and ORF mRNA abundance (RPKM of mRNA sequencing, ρ = 0.48) than with their TE's (*Figure 4B–C*). In contrast, the Gini score exhibits its highest correlation with TE (ρ = −0.76, *Figure 3C*) and is poorly correlated with mRNA abundance (ρ = −0.05) (*Figure 4—figure supplement 1C*). This suggests that codon bias may be evolutionarily selected to correspond to the ORF expression level rather than to its translation efficiency. Additionally, there is evidence that codon usage correlates with mRNA half-life in both eukaryotes and prokaryotes (*Boël et al., 2016*; *Presnyak et al., 2015*). ORF-wide codon usage (tAI) and intrinsic mRNA structure appear to be largely independent variables, as they show little correlation with each other (*Figure 4—figure supplement 1D*). Although a novel metric quantifying codon influence was highly successful at predicting protein production from overexpressed

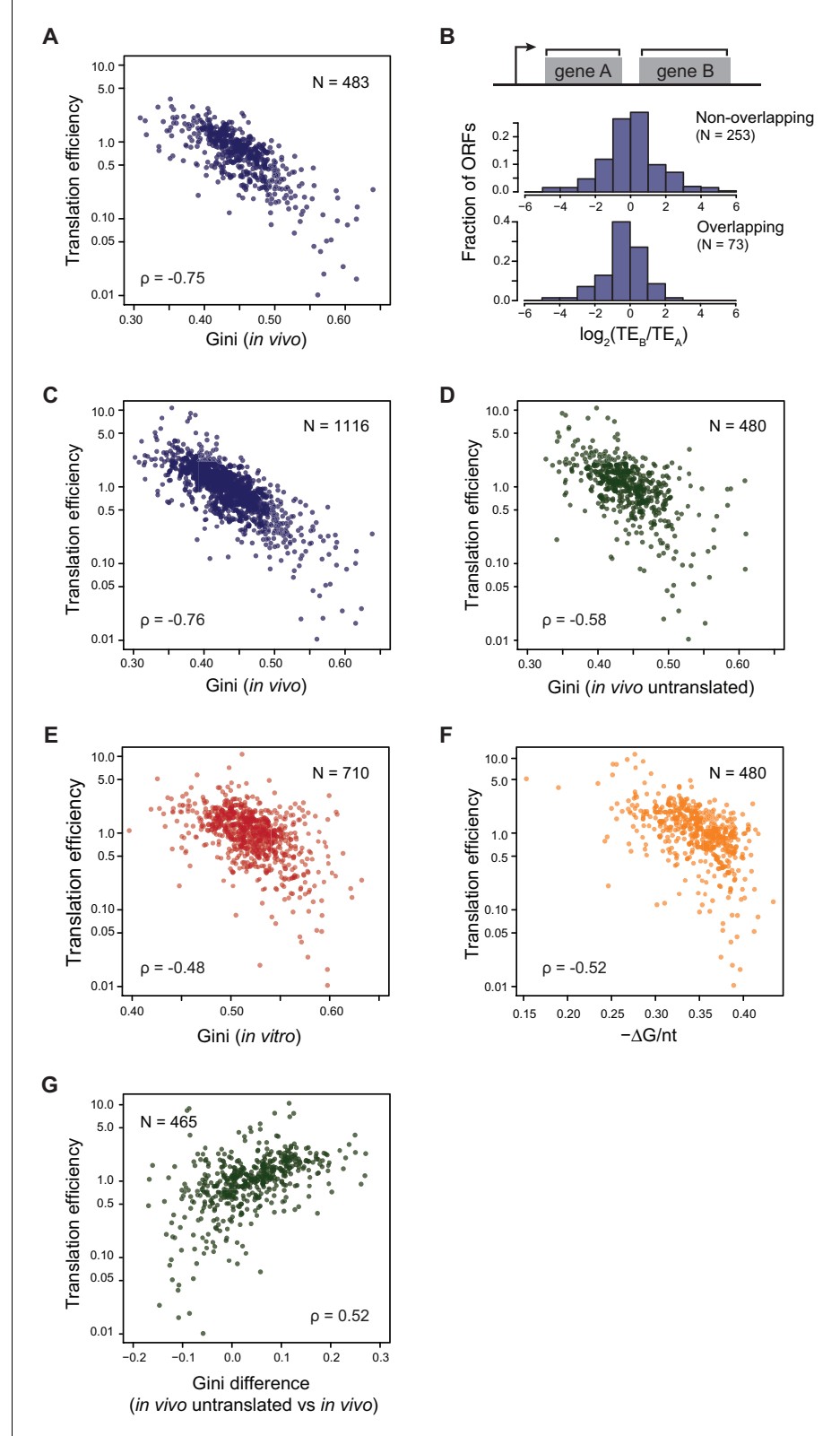

**Figure 3.** Translational efficiency (TE) is highly correlated with ORF mRNA structure. (**A**) Plots comparing the Gini indices of ORFs in polycistronic operons calculated from in vivo DMS-seq to their TEs (N = 483). (**B**) Histograms of TE ratios between adjacent non-overlapping (N = 253) or overlapping (N = 73) ORFs in operons (see Materials and methods for details). Overlapping ORFs are ORF pairs for which the annotated stop codon of the

*Figure 3 continued*

upstream ORF overlaps or is 3′ of the start codon of the downstream ORF. (**C–E**) Plots comparing the Gini indices of endogenous ORF mRNAs calculated from DMS-seq data of: C. in vivo RNA; D. in vivo RNA with no translating ribosomes (Ksg treated cells); E. in vitro modified refolded mRNA, to their TEs. For this and all subsequent panels, data calculated using different sets of ORFs are summarized in *Supplementary file 1–3*. (**F**) Plot comparing computationally predicted mRNA structure (- minimum free energy / nucleotide; -ΔG/nt) of the entire ORF body to TE. (**G**) Plots of the difference in the Gini index between untranslated (ksg-treated) and translated in vivo mRNA against their TE for the 465 ORFs in both datasets. X-axis: Gini index (in vivo untranslated) – Gini index (in vivo), normalized by the average of the two.

The following figure supplement is available for figure 3:

**Figure supplement 1.** Correlation between the mRNA structural level and translation efficiency.

exogenous genes transcribed by T7 RNA polymerase (*Boël et al., 2016*), it is relatively weakly correlated with TE under physiological conditions for endogenous genes (ρ = 0.29) (*Figure 4D*). This suggests that the codons providing efficient translation of an over-expressed transgene may differ from the efficient codons for an endogenous gene, as overexpression causes amino acid starvation and concomitant alteration of charged tRNA pools (*Plotkin and Kudla, 2011*; *Welch et al., 2009*; *Dittmar et al., 2005*; *Elf et al., 2003*).

Overall, ORF-wide mRNA secondary structure is by far the strongest and most significant predictor of endogenous TE compared to the other factors discussed above (*Figure 4—figure supplement 2*). A linear regression model that includes the addition of the Boel metric, tAI, and Shine-Dalgarno sequence strength showed marginal improvement in the predictive power compared to the ORF-wide structure alone (Figure 4—source data 1). Therefore, rather than being a driver for TE, codon optimization may be critical for highly expressed genes due to higher demand for these tRNAs and may play a role in setting the appropriate mRNA half-life.

## Overexpressing a protein with a rare codon alters endogenous translation

Our results thus far indicate that the rules for endogenous translation differ from those for overexpressed genes, particularly in the role of codon choice. Considering the fact that the expression of each tRNA species is tuned to the endogenous usage of its cognate codon(s) (*Dong et al., 1996*), overexpressed transgenes are likely to perturb the balance between codon usage and tRNA abundance, creating a global translation defect (*Shah et al., 2013*). To directly test this hypothesis, we evaluated the effects of transgene overexpression containing one codon at a time. We constructed a synthetic gene with only one sense codon after the initiating codon and expressed this minimal ORF to directly assess the influence of a single tRNA and amino acid without additional complications from the protein product. When the minimal ORF contains the rare leucine codon CUA, which has only one cognate tRNA, we observed elevated ribosome occupancy at CUA codons in endogenous genes (*Figure 4G*). In particular, slow translation at CUA codons in the *leuL* leader sequence triggers the expression of leucine biosynthetic genes (*Figure 4H*), whereas overexpressing the minimal ORF with the common leucine codon CUG or without any coding sequence does not change the expression level or ribosome occupancy at leucine codons of endogenous genes (*Figure 4G and I*). These results suggest that overexpression of a rare codon and not a common codon can deplete the pool of free cognate aa-tRNA molecules, leading to global perturbation of translation. Cells expressing a transgene that contain more rare codons are thus under a different physiological state compared to WT cells solely expressing endogenous genes.

## mRNA structure at ORF boundaries in a polycistronic operon

Bacterial operons are densely packed with ORFs, as the majority of adjacent ORFs (62%) are separated by only 25nt or less (*Figure 5A*). Our finding that ORF mRNAs have a roughly similar degree of structure (Gini index) throughout their entire length (*Figure 2A–D*), but that the degree of structure of adjacent ORF mRNAs on polycistronic transcripts can differ significantly (*Figure 2G*) suggests that mRNA structure undergoes a sharp transition at ORF boundaries.

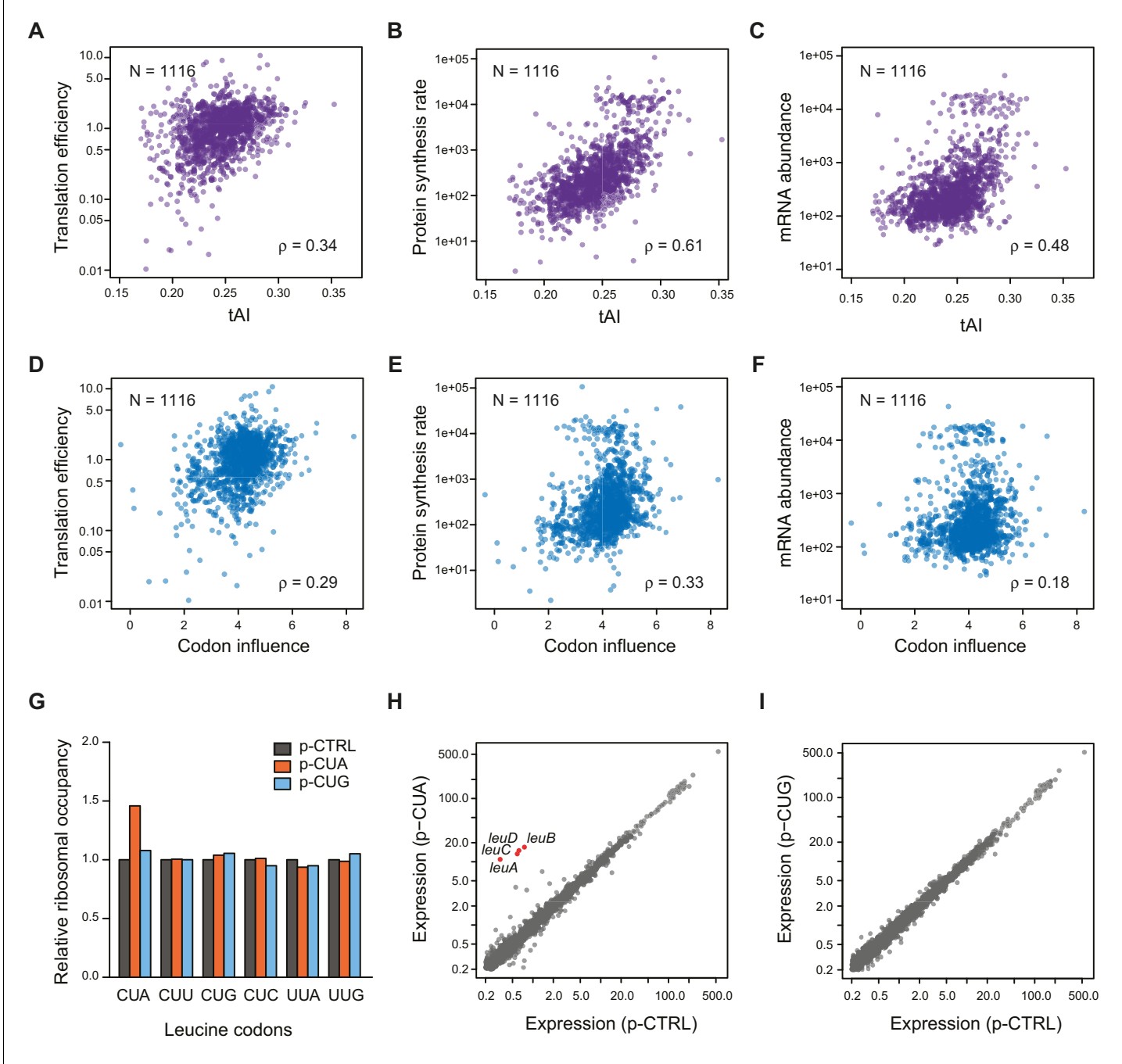

**Figure 4.** Correlation of Other mRNA features with TE. (**A–C**) Plots comparing tAI (tRNA adaptation index) of the entire ORF against: A. translation efficiency (TE, protein synthesis rate per mRNA); B. protein synthesis rate (average ribosome footprint density); C. mRNA abundance (RPKM mRNA sequencing) of the ORF. For this and the following panels of this figure, the 1116 ORFs in the in vivo RNA DMS-seq dataset are analyzed (*Supplementary file 1*). (**D–F**) Plots comparing codon influence across the entire ORF defined from overexpressing exogenous genes (*Boël et al., 2016*) against: D. translation efficiency; E. protein synthesis rate; F. mRNA abundance of the ORF. (**G**) Average ribosome occupancy at leucine codons in endogeneous genes when overexpressing a control plasmid (p-CTRL without a mini ORF) or plasmids with a heterologous CUA mini-ORF (p-CUA) or a CUG mini-ORF (p-CUG). The ribosome occupancy at each leucine codon was normalized by the average ribosome density of the ORF. The relative ribosome occupancy of that specific leucine codon was averaged across ORFs and normalized to that of the cells with control plasmid. (**H–I**) Gene expression changes with the control plasmid and heterologous overexpression of CUA codon mini-ORF (**H**) or CUG codon mini-ORF (**I**). The average ribosome footprint density of individual genes (see Materials and methods for details) was plotted in log2 scale.

The following source data and figure supplements are available for figure 4:

*Figure 4 continued on next page*

*Figure 4 continued*

**Source data 1.** Linear regression model to predict TE based on different mRNA features.
**Figure supplement 1.** Effect of SD strength, tAI, and codon influence on predicting TE of endogenous genes.
**Figure supplement 2.** Comparison of the relative significance of different mRNA features in predicting TE.

We examined the structural organization of mRNA at ORF boundaries in polycistronic mRNAs. We find that the local degree of mRNA folding immediately downstream of the start site correlates with the TE of the downstream gene, but that this correlation rapidly diminishes upstream of the start site. Conversely, local mRNA structure upstream of the start site is only correlated with the TE of the upstream ORF (*Figure 5B*). This is true not only for mRNAs that are being translated (WT cells; *Figure 5B*) but also for untranslated mRNAs (kasugamycin-treated cells; *Figure 5C*), in vitro refolded mRNAs (*Figure 5D*) and computationally predicted mRNA structures (*Figure 5—figure supplement 1A*). Thus, mRNA structure undergoes a sharp transition at ORF boundaries, and polycistronic mRNAs consist of distinct ORF-length structural domains.

## ORFs are isolated from each other by forming ORF-specific RNA structures

The close packing of ORF mRNAs raises the issue of how they maintain distinct structural domains, and suggests that bacterial ORFs may be marked not only by start and stop codons, but also by features that assist within-ORF mRNA folding. To investigate this, we computationally predicted the structure of mRNA extending −250 to +250 nt from the translation start at the boundary of adjacent ORF pairs within the same operon. Because folding algorithms often predict a large ensemble of possible folds for a long stretch of RNA, we constrained the predictions by forcing positions that were highly DMS-modified to be unpaired in the predicted structures.

Consistent with previous studies (*Eyre-Walker and Bulmer, 1993*; *Scharff et al., 2011*; *Bentele et al., 2013*; *Del Campo et al., 2015*), we found a lack of structure in the immediate vicinity of the start sites for most ORFs (*Figure 5E*). Downstream from this structure-free zone (25–50 nt), endogenous mRNA has a high propensity to base pair with regions further downstream, that is pairing within the same ORF (*Figure 5F*). Conversely, nucleotides located 25-50nt upstream of the start site have a strong preference for interacting with regions further upstream in the preceding ORF (*Figure 5F*). Importantly, in vivo mRNA without translating ribosomes and in vitro probed mRNA (*Figure 5E–F*) also showed such preferences. Thus, the sharp transition in the directionality of base-pairing around start sites is driven by the mRNA sequence itself, promoting ORF-centric units of secondary structure.

We experimentally investigated the effects of disrupting a region that promotes independent mRNA folding within adjacent ORFs. The *dusB-fis* operon is composed of a highly structured upstream gene (*dusB*) and a poorly structured downstream gene (*fis*) separated by 25 nucleotides. The two ORFs have an ~100 fold difference in TE (*Figure 6A*). Previous work indicated that the upstream *dusB* gene has a stem-loop structure near the 3' end of the gene; that mutationally disrupting the stem-loop (Mutation M3; *Figure 6A*) decreased translation of *fis;* and that restoring base pairing by a second mutation (M2) restored *fis* translation for unknown reasons (*Nafissi et al., 2012*). After confirming these results (*Figure 6B*), we performed DMS-seq on WT and mutant cells to determine whether destroying the stem-loop decreased *fis* translation by reducing the structural isolation of *dusB* and *fis*.

A model of the structure of the *dusB-fis* interface constrained by DMS-seq data (*Figure 6D*) indicates that the *dusB* and *fis* ORFs are structurally distinct in WT and double mutant (M3/M2 or M3:2) cells (*Figure 6E*), but that M3 increases the structure of *fis* mRNA (*Figure 6C* and *Figure 6—figure supplement 1*). In the M3 mutant, the -58 ~ −53nt region (blue) pairs with the +9 ~ +14 nt region of *fis* (red), rather than forming a stem-loop structure within *dusB* as it does in WT and M3:2 cells (*Figure 6E–F*). The increased structure of *fis* mRNA in the M3 mutant starts at the ORF boundary and spreads across the entire downstream coding region of *fis* (*Figure 6—figure supplement 1*).

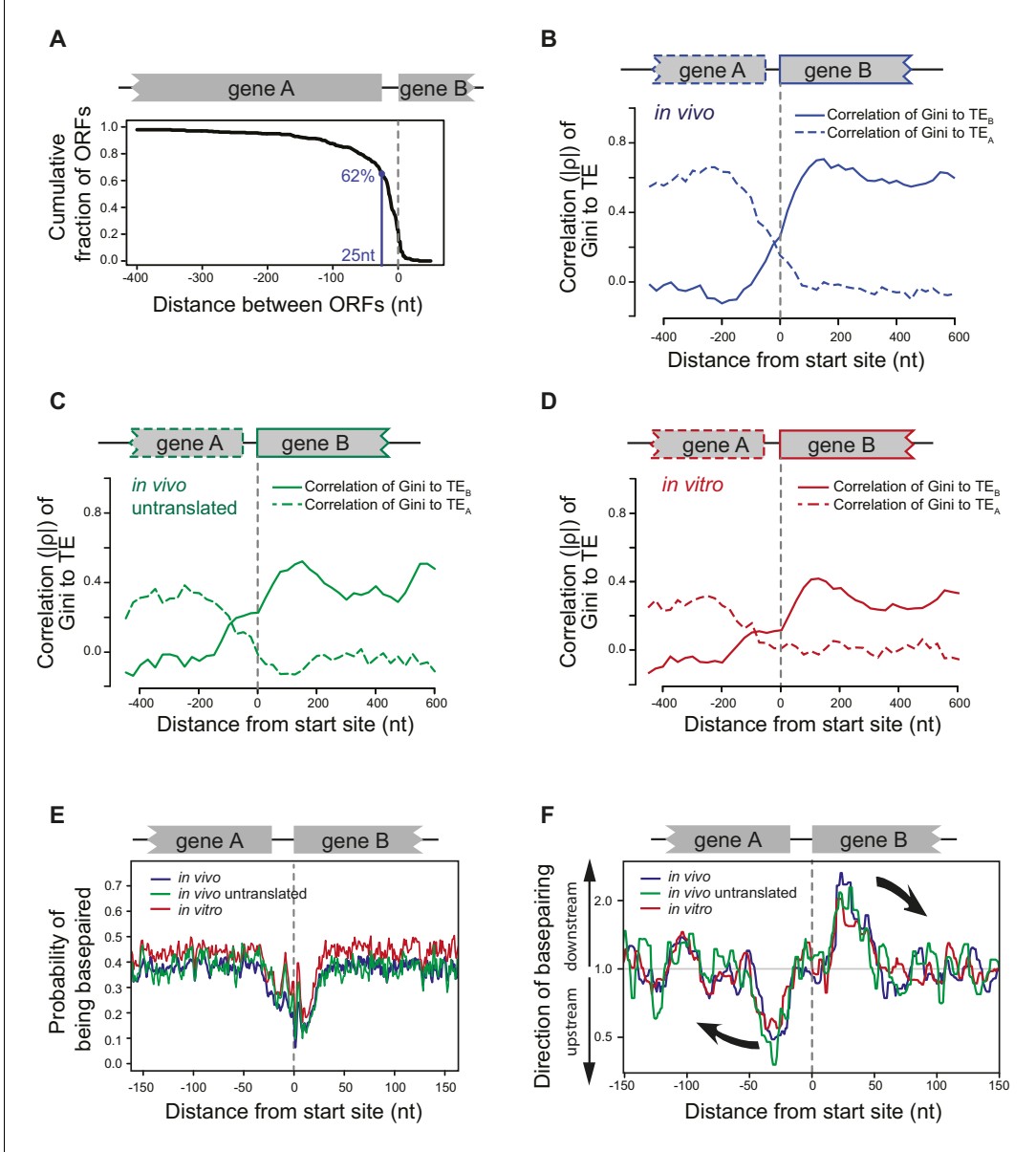

**Figure 5.** ORFs are isolated from each other by forming ORF-specific RNA structures. (**A**) Cumulative distribution of spacing between adjacent ORFs within operons of *E. coli*. X-axis: distance from 3' of the stop codon of upstream genes (gene A) to 5' of the start codon of downstream genes (gene B). (**B–D**) Correlation between local mRNA structure quantified by Gini index and TE of adjacent ORFs in the same operon. X-axis: distance from the 5' of start codon of downstream ORFs (gene B). Y-axis: the absolute value of correlation (Spearman's ρ) of local Gini indices, calculated from DMS-seq of in vivo mRNA (**B**), in vivo untranslated mRNA (ksg-treated) (**C**) or in vitro modified mRNA (**D**), against TE of the upstream (gene A; dashed line) or the downstream (gene B; solid line) gene. Gini indices were calculated within 300 nt windows scanning across the boundary between adjacent ORFs within operons. The correlation to TE is plotted at the center of each 300 nt window. (**E**) Meta-gene analysis of mRNA structure in the vicinity of translation initiation sites. Structure was predicted by applying the DMS-seq constrained minimum free-energy model calculated from in vivo mRNA (blue), in vivo untranslated mRNA (ksg-treated; green) or in vitro modified mRNA (red). Mean predicted base-pairing probability of each nucleotide (averaged across genes) was plotted across the boundary between adjacent ORFs within operons. (**F**) Plot of directionality of RNA folding at ORF boundaries. At each position, the probability of base pairing with every other position was calculated for each ORF examined. The average sum probability of base-pairing with any nucleotide in a 60 nt window upstream and in a 60 nt window downstream was calculated. Y-axis: the ratio of the downstream base-pairing probability to the upstream base-pairing probability at each position (X-axis). The black arrows indicate preferential folding direction.

The following figure supplement is available for figure 5:

**Figure supplement 1.** Structural isolation between mRNA of adjacent ORFs on the same operons.

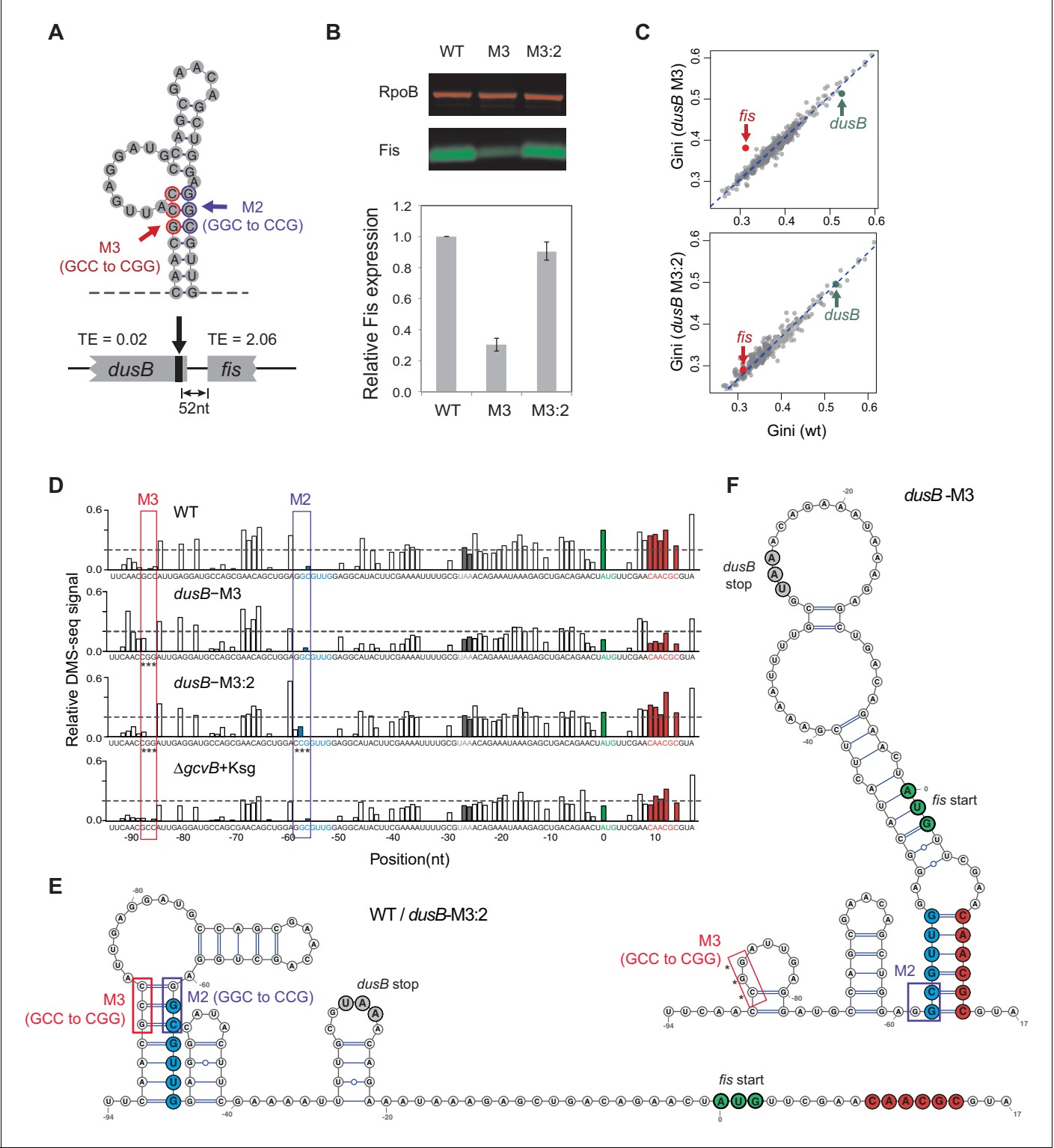

**Figure 6.** Disruption of structural isolation between *dusB* and *fis* affects *fis* translation. (**A**) mRNA structure at the 3' end of *dusB*, with mutations M3 and M2 indicated. Translation efficiencies (TEs) of *dusB* and *fis* in WT cells are 0.02 and 2.06, respectively. (**B**) The *dusB*-M3 mutation decreases Fis expression and is rescued by the complementary M2 mutation. Western blot compares Fis protein amounts in WT, *dusB*-M3 and *dusB*-M3:2 double mutant cells, with RpoB protein as an internal control. (**C**) Scatter plots comparing Gini indices of ORFs in WT cells to those in *dusB*-M3 or in *dusB*-M3:2 double mutant cells. Outlier test: *fis*, Bonferonni p-value=1.02e−05 (*dusB*-M3); p-value>0.05 (*dusB*-M3:2). (**D**) Normalized DMS-seq signals at the *dusB*-*fis* boundary region from different samples as indicated. Positions of M3 and M2 are highlighted, with asterisks indicating mutated nucleotides. X-axis:
*Figure 6 continued on next page*

*Figure 6 continued*

distance from 5' end of the *fis* start codon. Y-axis: normalized DMS-seq signals. Dashed line: threshold (0.2) above which the A/C bases are predicted to be unpaired (see Materials and methods). (E) mRNA structure at the *dusB-fis* boundary region of WT or *dusB*-M3:2 cells, predicated by constraining a minimum free-energy model with DMS-seq measurements. Locations of mutations M3 and M2 are as indicated. (F) mRNA structure at the *dusB-fis* boundary region of *dusB*-M3 mutant cells, predicated by constraining a minimum free-energy model with DMS-seq measurements. CGG residues labeled with asterisks indicate the M3 mutation.

The following figure supplement is available for figure 6:

**Figure supplement 1.** mRNA secondary structure at the *dusB-fis* boundary region of WT cells and *dusB* mutants.

Thus, mutation M3 induces long-range interactions between mRNA of the *dusB* and *fis* ORFs, which are normally structurally insulated from each other.

In *toto*, our results suggest that specific sequences isolate mRNA folding within adjacent ORFs thereby minimizing structural crosstalk between adjacent ORFs. Disruption of structural boundaries affects both local and long-range mRNA folding, which is likely to be critical for programming the degree of translational isolation between ORFs on the same mRNA.

## Discussion

Translation is a highly controlled process in bacteria, making it critical to understand the mRNA features contributing to differential translatability. Numerous studies have investigated the important question of which features control protein production from overexpressed, foreign ORF mRNAs, identifying codon usage and local structure around the translation start site as key variables. However, these studies have left open the question of which mRNA features regulate endogenous translation. The importance of this question is highlighted by the observation that the rate of protein production from each ORF in a polycistronic mRNA can vary as much as 100-fold. Our global study now examines this issue. Our principal finding is that ORF mRNAs have modular structures within polycistronic mRNAs and that ORF-wide mRNA structure rather than codon usage correlates most strongly with the translation efficiency of endogenous ORFs.

Our analysis of mRNA structure revealed the unanticipated finding that operonic mRNAs have modular structures. Each ORF mRNA in the operon has a characteristic degree of structure, with highly correlated Gini scores between their first and second halves. This correlation persists in the absence of translation, when mRNAs are refolded in vitro and when structure is determined computationally (*Figure 2A–D*). In stark contrast, there is little correlation between the extent of structure in adjacent ORFs (*Figure 2G*). Additionally, and consistent with earlier computational findings, we observe a small ~25 nucleotide region beginning at the translation start that is more unstructured than the remainder of the ORF (*Figure 5E*). Thus, polycistronic mRNAs consist of a series of ORF-wide modules each with characteristic but different extents of structure, punctuated by regions of low basepairing at the translation start site (*Figure 7*). Maintenance of a common degree of structure throughout an ORF suggests that this parameter, like reduced structure at the start of ORFs, is a selective force in the evolution of ORF sequence, providing yet another constraint on mRNA sequence beyond codon adaptation (*Sharp and Li, 1987*).

We find that the TE of each ORF correlates very highly, and most strongly with the ORF-wide extent of mRNA structure. We have begun to deconvolute the 'chicken and egg' problem of whether mRNA structure is a cause or a consequence of translation by examining the correlation of TE to ORF-wide structure when translation is inhibited. This removes the ribosome contribution but retains vectorial folding, RNA binding proteins and in vivo concentrations of salts and macromolecules. Untranslated mRNA structure is highly correlated with TE but less so than translated mRNA (*Figure 3C–D*). Moreover, the difference in the mRNA structure of an ORF with and without translation is highly correlated to its TE (*Figure 3G*). Thus, poorly translated mRNAs, have virtually identical extents of structure with and without translation, but more highly translated RNAs become increasingly more unstructured. Finally, computationally predicted structures or those obtained from in vitro refolded mRNAs correlate somewhat more poorly with TE ($\rho = -0.52$ or $-0.48$ respectively) than

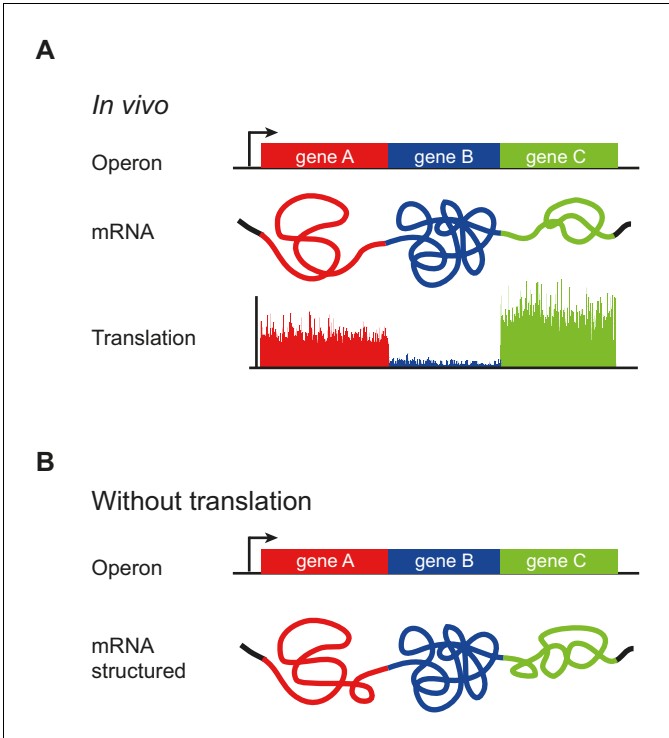

**Figure 7.** Model of operon mRNA structural organization. Polycistronic mRNAs are organized into ORF-centric modules with characteristic but different extents of mRNA structure, punctuated by regions of low basepairing close to the translational start site (**A**). The intrinsic ORF-wide mRNA structure is highly predictive of translation efficiency (**B**), and is amplified by translation, in a self-reinforcing loop, to provide the mRNA structure that ultimately specifies the translation of each ORF in an operon.

the structure of untranslated in vivo mRNA ($\rho = -0.58$) (**Figure 3D–F**). This suggests that features of the in vivo cell, besides translation by the ribosome, may also affect mRNA structure.

Taken together, these results suggest that the intrinsic mRNA sequence itself encodes a rough blueprint for the ORF-centric mRNA structures that are predictive of TE. These structures are then amplified by translation and other features of the living cell, in a self-reinforcing loop, to provide the structure that ultimately specifies the translation of each ORF.

Interestingly, *E. coli* has chosen to insert a predominance of its low TE ORFs into operons where adjacent genes have moderately high TE's (for example, the *rpsF-priB-rpsR-rplI* operon shown in **Figure 3—figure supplement 1A**). All of its 10 lowest TE ORFs, and 86% of its ORFs in the bottom 10% of TEs are located in operons, compared with 58% of all ORFs. The evolutionary advantage of this arrangement is not known, but may relate either to decreasing transcriptional noise or to mRNA stabilization.

The necessity for achieving widely different TEs for adjacent ORFs in operons may have driven the evolution of the ORF-centric mRNA folding strategy. As the translation termination codon of most ORFs is separated by less than 25nt of untranslated mRNA from the start site of the downstream ORF, the abundant ribosomes of the highly translated ORF could transiently open the structure of the poorly translated ORF and increase the accessibility of its start site. The propensity for in-ORF mRNA folding at both the beginning and ends of ORFs may prevent the upstream ORF from influencing the structure and hence TE of the downstream ORF, effectively insulating each ORF from its neighbors. We have identified small regions, located about 25–50 nucleotides both downstream and upstream of ORF translation start sites that preferentially pair within their ORFs. These regions may reinforce the folding barriers between adjacent ORF mRNAs, as we demonstrated for the *dusB-fis* operon. Interestingly, RNA polymerase pausing is enriched at translation start sites (**Larson et al.,**

*2014*) and this may reinforce ORF-centric structural insulation by allowing ORFs to fold independently during the pioneer round of translation.

It is likely that the extent to which adjacent ORFs are insulated has been tuned. Approximately 15% of ORF pairs have overlapping stop and start codons and translational coupling has been demonstrated in some cases (*Aksoy et al., 1984*; *Oppenheim and Yanofsky, 1980*; *Schümperli et al., 1982*; *Yates and Nomura, 1981*). This overlap may enable upstream ribosomes to influence downstream ORF translation by unwinding mRNA structure, thereby promoting translational coupling. Indeed, it is likely that the propensity for in-ORF basepairing is slightly weaker for overlapping ORF pairs than for non-overlapping ORF pairs (*Figure 5—figure supplement 1B–C*).

The precise role of modular ORF structures that provide a rough blueprint for TE has not yet been established. It is certainly possible that the mRNA structure of an entire or a significant fraction of the ORF is required to define translation initiation, as has been demonstrated experimentally for *rimM* (*Wikström et al., 1992*). Alternatively, a constant degree of ORF-wide mRNA structure may be the most robust way to ensure the appropriate amount of mRNA structure around the translation initiation site. In support of this idea, a recent study using in situ codon mutation of the *E. coli* essential gene *infA* showed that mutations of codons even far downstream from the start of the gene can be deleterious if they disrupt the native 5' RNA conformation via long-range structural interactions predicted computationally (*Kelsic et al., 2016*). ORF-wide structures may also play additional roles. For high TE (poorly structured) ORFs, extended lack of structure may provide the landing pad necessary to capture a large pool of non-specifically bound 30S subunits to wait for opening of the SD and start codon, the so-called 'standby model' of translation initiation (*Adhin and van Duin, 1990*; *de Smit and van Duin, 2003*). Additionally, the ORF-centric mRNA folding strategy may have been driven by the necessity for adjacent ORFs to have discrete, often significantly different TEs. Finally, ORF-wide mRNA structures may help set the rate of endonucleolytic cleavage. The function of these modules is an important area for future inquiry.

Although the TE of endogenous ORFs is primarily predicted by the extent of its mRNA structure, translatability of overexpressed foreign ORFs appears to be strongly driven by codon usage and tRNA limitation. This difference may arise from the fact that codon usage and tRNA abundance are largely balanced under physiological conditions, but become imbalanced when foreign ORFs are overexpressed, and we have directly demonstrated that this is the case (*Figure 4G–I*). This suggests that synthetic biologists and the cell tune translation in different ways. However, synthetic biologists struggle to robustly program differential translation of ORFs on the same mRNA. Our finding that polycistronic mRNAs consist of ORF-wide modules with set amounts of structure that are insulated from their neighbors may be key to this issue. Design approaches that incorporate appropriate mRNA structures may have the potential to produce the finely tuned synthesis rates observed in natural operons.

# Materials and methods

## Strains and growth conditions

*E. coli* K-12 MG1655 (RRID:SCR_002433) was used as the WT strain. All culture experiments were performed in MOPS medium supplemented with 0.2% glucose, all amino acids except methionine, vitamins, bases and micronutrients (Teknova, Hollister CA). Cells were grown in an overnight liquid culture at 37°C, diluted to an $OD_{420}$ = 0.001 in fresh medium and grown until $OD_{420}$ reached 0.4 where samples were collected. Multiple deletion strains were generated by transduction of FRT-flanked deletion alleles from the Keio collection (*Baba et al., 2006*) followed by marker excision by Flp recombinase (*Cherepanov and Wackernagel, 1995*). All major experiments were biologically repeated for at least twice (see raw data files for sequencing data).

In the experiment testing the effects of overexpressing the rare CUA leucine codon and the common CUG leucine codon, plasmids with pBR322 origin of replication was constructed to have a mini ORF ATGCTATAA or ATGCTGTAA driven by an IPTG-inducible promoter. The plasmid also contains lacI$^q$ to increase the expression of lac repressor. MG1655 containing the control plasmid (without mini ORFs) and MG1655 containing the plasmid with CUA or CUG mini ORF were grown overnight in MOPS rich glucose medium with 100 µg/ml ampicillin, diluted 1:1000 into 250 ml pre-warmed fresh medium containing 1 mM IPTG next morning. Cells were grown at 200 rpm at 37°C and harvested when $OD_{600}$ reached 0.3 by vacuum filtration.

## Ribosome profiling sample capture

The protocol for bacterial ribosome profiling with flash freezing was described (*Li et al., 2014*). Briefly, 200 mL of cell culture were filtered rapidly and the resulting cell pellet was flash-frozen in liquid nitrogen and combined with 650 µL of frozen lysis buffer (10 mM $MgCl_2$, 100 mM $NH_4Cl$, 20 mM Tris-HCl pH 8.0, 0.1% Nonidet P40, 0.4% Triton X-100, 100 U/mL DNase I (Roche, St. Louis MO), 1 mM chloramphenicol). Cells were pulverized in 10 mL canisters pre-chilled in liquid nitrogen. Lysate containing 0.5 mg of RNA was digested for 1 hr with 750 U of micrococcal nuclease (Roche) at 25°C. The ribosome-protected RNA fragments were isolated using a sucrose gradient followed by hot acid phenol extraction. Library generation was performed using the previously described strategy (*Li et al., 2014*) detailed below.

## Total mRNA sample capture

For experiments performed in parallel with ribosome profiling, total RNA was phenol extracted from the same lysate that was used for ribosome footprinting. For experiments performed independently of ribosome profilng experiments, and for total mRNA used for in vitro DMS-seq experiments, 4 mL of $OD_{420}$ = 0.4 culture was added to 500 µL of ice-cold stop solution (475 µL of 100% EtOH and 25 µL acid phenol), vortexed, spun for 2 min at 8000 rpm, and the cell pellet was flash frozen in liquid nitrogen. Total RNA was then hot acid phenol extracted. For mRNA-seq experiments, ribosomal RNA and small RNA were removed from the total RNA with MICROBExpress (Ambion, Grand Island NY) or Ribozero (Epicenter, Madison WI) and MEGAclear (Ambion), respectively. mRNA was randomly fragmented as described (*Ingolia et al., 2009*). The fragmented mRNA sample was converted to a complementary DNA library with the same strategy as for ribosome footprints.

## Library generation for ribosome profiling and mRNA-seq

The footprints and mRNA fragments were ligated to miRNA cloning linker-1 (IDT) 5rApp/CTGTAGG-CACCATCAAT/3ddC/, using a recombinantly expressed truncated T4 RNA ligase 2 K227Q produced in our laboratory. The ligated RNA fragments were reverse transcribed using the primer 5'/5Phos/GATCGTCGGACTGTAGAACTCTGAACCTGTCGGTGGTCGCC GTATCATT/iSp18/CACTCA/iSp18/CAAGCAGAAGACGGCATACGAATTGATG GTGCCTACAG 3'. The resulting cDNA was circularized with CircLigase (Epicentre), and PCR amplification was done as described previously (*Ingolia et al., 2009*).

## DMS modification

For in vivo DMS modification, 15 mL of exponentially growing *E. coli* were incubated with 750 µL DMS. Incubation was performed for 2 min at 37°C. For kasugamycin (ksg) experiments, ksg was added to a final concentration of 10 mg/mL to *ΔgcvB* cells for 2 min at 37°C prior to DMS modification. Untreated *ΔgcvB* cells were also modified by DMS and collected in parallel as control. DMS was quenched by adding 30 mL 0°C stop solution (30% $\beta$-mercaptoethanol, 25% isoamyl alcohol), after which cells were quickly put on ice, collected by centrifugation at 8000 *g* and 4°C for 2 min, and washed with 8 mL 30% BME solution. Cells were then resuspended in 450 µL total RNA lysis buffer (10 mM EDTA, 50 mM sodium acetate pH 5.5), and total RNA was purified with hot acid phenol (Ambion). For in vitro DMS modifications, mRNA was collected from cells that were not treated with DMS. Two micrograms of mRNA was denatured at 95°C for 2 min, cooled on ice and refolded in 90 µL RNA folding buffer (10 mM Tris pH 8.0, 100 mM NaCl, 6 mM $MgCl_2$) at 37°C for 30 min then incubated in either. 2% DMS for 1 min (95°C) or 4% DMS for 5 min (37°C). The DMS reaction was quenched using 30% BME, 0.3 M sodium acetate pH 5.5 and precipitated with 2 µL GlycoBlue and 1X volume of isopropanol.

## Library generation for DMS-seq samples

Sequencing libraries were prepared as described (*Rouskin et al., 2014*). Specifically, DMS treated mRNA samples were denatured for 2 min at 95°C and fragmented at 95°C for 2 min in 1x RNA fragmentation buffer ($Zn^{2+}$ based, Ambion). The reaction was stopped by adding 1/10 vol of 10X Stop solution (Ambion) and quickly placed on ice. The fragmented RNA was run on a 10% TBU (Tris borate urea) gel for 60 min. Fragments of 60–70 nucleotides in size were visualized by blue light (Invitrogen, Carlsbad CA) and excised. Reverse transcription was performed in a 20 µL volume at

52°C using Superscript III (Invitrogen), and truncated reverse transcription products of 25–45 nucleotides were extracted by gel purification.

## Sequencing

Sequencing was performed on an Illumina HiSeq 2000 or 4000. Sequence alignment with Bowtie v. 0.12.0 mapped the footprint data to the reference genomes NC_000913.fna obtained from the NCBI Reference Sequence Bank. Sequencing data from mutated strains were aligned to appropriately modified genome. For ribosome footprint and mRNA-seq samples, the center residues that were at least 12 nucleotides from either end were given a score of 1/N in which N equals the number of positions leftover after discarding the 5′ and 3′ ends. For DMS-seq samples, read counts were assigned to the base immediately 5′ of the 5′ end of each read, which is the base that was DMS-modified.

## Translation efficiency calculation

Data analysis was performed with custom scripts written for R 2.15.2 and Python 2.6.6. To calculate mRNA abundance, the number of mRNA sequencing reads mapped to a gene, following a Winsorization applied to trim the top and bottom 5% of reads, was divided by the length of the gene to yield the number of reads corresponding to the message per thousand bases of message per million sequencing reads (RPKM). The protein synthesis rate of individual ORFs was measured by average ribosome footprint density of the ORF calculated as described in (*Li et al., 2014*). First, genes with less than 128 reads mapped and genes with unconventional translation events were excluded from the analysis, which include (1) genes encoding selenoproteins (e.g. *fdhF*, *fdoG*, *fdnG*); (2) proteins with nearly identical coding sequences (e.g. *gadA* and *gadB*, *ynaE* and *ydfK*, *ldrA* and *ldrC*, *ybfD* and *yhhI*, *tfaR* and *tfaQ*, *rzoD* and *rzoR*, *pinR* and *pinQ*). Second, sequencing reads from ribosome profiling mapped to the first and last five codons of the gene were excluded to remove effects of translation and termination. Third, correction for the variations in translation elongation rate was done in three steps as described in *Li et al. (2014)*: (1) correcting for the elevated ribosome footprint density observed for the first 50–100 codons (*Oh et al., 2011*); (2) correcting for the elevated density at the ribosomal anti-Shine-Dalgarno (aSD) site (*Li et al., 2012*); (3) correcting for other possible ribosome pausing using 90% Winsorization, by removing the top and bottom 5% of the ribosome profiling signal for each gene. Finally, the average ribosome footprint density of a gene was calculated by dividing the corrected number of mapped ribosome footprint reads by the corrected length of the gene. Translation efficiency of a gene was calculated by normalizing the average ribosome footprint density by the mRNA abundance of the gene (defined above). The average ribosome footprint density (i.e. protein synthesis rate), mRNA abundance, and translation efficiency of genes from different samples are listed in *Supplementary file 1–4*.

## Computational prediction of RNA structures

For identification of unpaired bases, raw DMS-seq data was normalized to the most highly reactive residue after removing outliers by 95% Winsorisation (all data above the 95th percentile is set to the 95th percentile). Bases with DMS-seq signal greater that 20% of the signal on the most highly reactive residue (after normalization) were called 'unpaired'. For determination of *rimM* mRNA structures, a Viennafold (*Hofacker, 2003*) (http://rna.tbi.univie.ac.at/) minimum free-energy model of the indicated region was generated, constrained by bases experimentally determined to be unpaired in the indicated dataset. Color-coding by DMS signal was done using VARNA (http://varna.lri.fr/).

## Computing the agreement with ribosomal RNA

The secondary structure models for *E. coli* ribosomal RNAs were downloaded from Comparative RNA Website and Project database (http://www.rna.icmb.utexas.edu/DAT/3C/Structure/index.php). The crystal structure model was downloaded from Protein Data Bank (http://www.pdb.org, PDB entries 3I1M, 3I1N, 3I1O, and 3I1P). The solvent-accessible surface area was calculated in PyMOL, and DMS was modeled as a sphere with 2.5 Å radius (representing a conservative estimate for accessibility because DMS is a flat molecule). Accessible residues were defined as residues with solvent accessibility area of greater than 2 Å². Unpaired residues in DMS-seq data were identified as described above. True positive bases were defined as bases that are both unpaired in the secondary structure model and solvent-accessible in the crystal structure model. True negative bases were

defined as bases than are paired (A-U or C-G specifically) in the secondary structure model. Accuracy was calculated as the number of true positive bases plus the number of true negative bases divided by all tested bases.

## Calculation of Gini index on DMS-seq data

The R package 'ineq' (https://github.com/cran/ineq) was used to calculate Gini indices over As and Cs in the region specified for each experiment. For each DMS-seq sample, Gini indices were calculated only for genes that had greater than an average of 15 reads per nucleotide across the ORF. Genes with discontinuous mRNA-seq reads (due to an early termination event or an internal promoter, 1% of genes) were excluded from the analysis. Specifically, Gini indices were calculated on mRNA-seq data, and a cutoff was created based on two standard deviations from the mean. The Gini indices of genes from different samples were listed in *Supplementary file 1–4*.

## Identification of adjacent open reading frames on operons

Adjacent ORFs in annotated operons often have differing levels of mRNA-seq reads, suggesting that they are not always on the same mRNA molecule. To identify adjacent ORFs expressed as a single operon, we assessed mRNA-seq data for equivalent mean message level and for signal continuity, as described below. Equivalent mean message level was assessed by first determining the variability in mean mRNA-seq read density within individual ORFs. There is a single transcript that extends over the entire body of the large majority of ORFs, and so the variability in mean read density level in the first half of each ORF was compared to mean read density in the second half of each ORF, and the variability in this distribution was used to define a cut-off for ORFs on a single message. Adjacent ORFs that fell within a 2σ cutoff in mean level (calculated to be a 1.5-fold difference in mRNA level) were determined to have equivalent mRNA level and were then assessed for signal continuity. Signal continuity was assessed by first determining the distribution of read density in windows within messages. Gini index of mRNA-seq signal was calculated for all 80nt windows within ORF bodies, and the variability in this distribution was again used to define a cutoff for continuous mRNA regions. Gini index were then calculated for 80nt windows tiling the region between adjacent ORFs. Gene pairs that fell within a 2σ cutoff defined by the intra-ORF distribution were considered to be a pair of adjacent ORFs on a single message.

## Directionality of interaction predictions

To determine the directionality of mRNA base pairing at ORF boundaries, sequence from −250 to +250 nt relative to the translation start site of the downstream gene was extracted for each adjacent pair of ORFs. A Viennafold (*Hofacker, 2003*) (http://rna.tbi.univie.ac.at/) minimum free-energy model of each 500nt sequence was generated (constrained by DMS-seq data). The predicted probability of each base interacting with every other base in each mRNA structure model was then extracted from the Viennafold output. For each position, the probability of that position base pairing with any position within the upstream or downstream 60nt was then calculated. The ratio between summed upstream over downstream interaction probability across all mRNAs was then calculated for each position.

## Measurement of total protein synthesis

1 µC of Perkin Elmer EasyTag $^{35}$S labeled methionine (Product # NEG709A) was mixed with 5 µL 288 µmol unlabeled methionine and 24 µL media. At the time of capture, 900 µL of culture was added to methionine mix, and was labeled on a shaker for 1 min at 37°C. After labeling, 100 µL of ice-cold 50% trichloracetic acid (TCA) was added to the sample, which was vortexed and placed on ice for at least 20 min to allow precipitation. Samples were then counted by running 100 µL of sample through a 25 mm APFC glass fiber filter (Millipore APFC02500, Hainesport NJ) pre-wetted with 750 µL of 5% TCA on a vacuum stand, and washing three times with 750 µL 5% TCA and three times with 750 µL 100% ethanol. Filters were then placed in MP Ecolume scintillation fluid and counted.

## Shine-Dalgarno sequence strength calculation

We used the RBS Calculator established by Salis et al downloaded from http://www.github.com/hsa-lis/Ribosome-Binding-Site-Calculator-v1.0 to predict the strength of Shine-Dalgarno sequence.

## tAI calculation

The measurement of tAI (tRNA adaptation index) was adapted from the previous works (*Tuller et al., 2010*; *dos Reis et al., 2004*), which gauges the availability of tRNAs for each codon within a gene. tAI incorporates different efficiency weights of the wobble interactions between codons and anticodons, with $w_i$ is defined as the relative adaptiveness value of codon $i$ of a gene (*Tuller et al., 2010*). The final tAI of a gene is the geometric mean of all its codons as shown below.

$$tAI = (\prod_{k=1}^{L} \omega_{ik})^{1/L}$$

$i_k$ is the $k$th codon of the gene and L is the length of the gene (excluding start and stop codons).

## Western blotting

Wild-type, *dusB*-M3, and *dusB*-M3:2 cells were grown in MOPS rich medium at 37°C till log phase ($OD_{420}$ ~0.3). 1 mL cells were collected, resuspended in 30 µL SDS loading buffer, and boiled for 5 min. 10 µL of cell lysate was subject to Blot 12% Bis-Tris plus gel (ThermoFisher scientific, Grand Island NY). Proteins were transferred to a nitrocellulose membrane (BIO-RAD, Hercules CA). The membrane was first incubated with rabbit polyclonal anti-Fis antibody (a kind gift from Dr. Reid C. Johnson at UCLA) and mouse monoclonal anti-RNAP $\beta$ subunit antibody (abcam, Cambridge MA), and then incubated with goat anti-rabbit IgG IRDye 800CW and anti-mouse IgG IRDye 680RD secondary antibodies (LI-COR, Lincoln NE). The blots were visualized and quantified by an Odyssey imaging system (LI-COR). The amount of Fis protein in each sample was normalized against the amount of RNAP $\beta$ subunit in the same sample.

All the processed and raw datasets of sequencing experiments were uploaded to NCBI GEO database with accession number GSE77617.

## Acknowledgements

This research was supported by the Center for RNA Systems Biology (JSW), the Howard Hughes Medical Institute (JSW), the Helen Hay Whitney Foundation (GWL) and the National Institutes of Health (CAG, DHB, YZ R01GM036278 and R01GM057755, DHB T32GM8284 and T32 EB009383, GWL, K99GM105913).

## Additional information

### Funding

| Funder | Grant reference number | Author |
|---|---|---|
| National Institutes of Health | R01GM036278 | David H Burkhardt Yan Zhang Carol A Gross |
| National Institutes of Health | R01GM057755 | Carol A Gross David H Burkhardt Yan Zhang |
| National Institutes of Health | T32GM8284 | David H Burkhardt |
| National Institutes of Health | T32 EB009383 | David H Burkhardt |
| Helen Hay Whitney Foundation | | Gene-Wei Li |
| National Institutes of Health | K99GM105913 | Gene-Wei Li |
| Howard Hughes Medical Institute | | Jonathan S Weissman |
| Center for RNA Systems Biology | | Jonathan S Weissman |

The funders had no role in study design, data collection and interpretation, or the decision to submit the work for publication.

## Author contributions
DHB, SR, YZ, Data curation, Formal analysis, Validation, Investigation, Methodology, Writing—original draft, Writing—review and editing; G-WL, Conceptualization, Supervision, Funding acquisition, Investigation, Methodology, Writing—original draft, Writing—review and editing; JSW, CAG, Conceptualization, Supervision, Funding acquisition, Writing—original draft, Writing—review and editing

## Author ORCIDs
Yan Zhang, http://orcid.org/0000-0001-5440-1414
Jonathan S Weissman, http://orcid.org/0000-0003-2445-670X
Carol A Gross, http://orcid.org/0000-0002-5595-9732

# Additional files

## Supplementary files
• Supplementary file 1. In vivo data summary. Data summary of ORFs that have ≥15 reads per nucleotide for in vivo DMS-seq (N = 1116).

• Supplementary file 2. In vitro data summary. Data summary of ORFs that have ≥15 reads per nucleotide for in vitro DMS-seq (N = 710).

• Supplementary file 3. In vivo untranslated (ksg-treated) data summary. Data summary of ORFs that have ≥15 reads per nucleotide for DMS-seq of in vivo untranslated (ksg-treated) mRNA (N = 480).

• Supplementary file 4. Common ORFs data summary. Data summary of ORFs that have ≥15 reads per nucleotide in all three conditions: DMS-seq of in vivo mRNA, in vivo untranslated mRNA and in vitro modified mRNA (N = 421).

## Major datasets
The following dataset was generated:

| Author(s) | Year | Dataset title | Dataset URL | Database, license, and accessibility information |
|---|---|---|---|---|
| Burkhardt DH, Rouskin S, Zhang Y, Li G, Weissman JS, Gross CA | 2016 | Operon mRNAs are organized into ORF-centric structures that predict translation efficiency | https://www.ncbi.nlm.nih.gov/geo/query/acc.cgi?acc=GSE77617 | Publicly available at the NCBI Gene Expression Omnibus (accession no: GSE77617) |

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
