## [Decision Letter]

Thank you for submitting your article "Operon mRNAs are organized into ORF-centric structures that influence translation efficiency" for consideration by *eLife*. Your article has been favorably evaluated by Detlef Weigel (Senior Editor) and three reviewers, one of whom is a member of our Board of Reviewing Editors. The reviewers have opted to remain anonymous.

The reviewers have discussed the reviews with one another and the Reviewing Editor has drafted this decision to help you prepare a revised submission. We hope you will be able to submit the revised version within two months.

The three reviewers of this manuscript are generally enthusiastic about the data and the potential explanation that mRNA structure in bacteria occurs in modules or blocks that correspond to ORFs, and that these structural units dictate to some significant extent TE. Despite overall enthusiasm, there are several key statistical issues that need to be addressed (see reviewer 2 comments) including an evaluation of the statistical significance of the comparisons in correlations (which were done for different sets of genes). The reviewers also agreed in consultation that the ideal approach to take would be to perform a multiple regression analysis (reviewer 2, point 5) to explain TE using the various parameters (mRNA structure, codon bias, SD sequence, structure around start codons, etc.). These statistical analyses should be reasonably straightforward and would significantly increase the impact of the manuscript.

*Reviewer #1:*

This manuscript seeks to explain how the translational efficiency of *E. coli* genes within operons can vary as much as 100-fold even though they are found on a single polycistronic mRNA. Using a genome-wide RNA structural probing method (DMS-Seq) the authors have made a major discovery, that mRNA structure occurs in modules or blocks corresponding to ORFs, where adjacent ORFs often have very different levels of structure. mRNA structure is strongly anti-correlated with translational efficiency. While they offer a sophisticated discussion about the role of translating ribosomes in opening up mRNA structure, they argue that the structure is to a large extent encoded within the mRNA sequence itself on the basis of structural probing experiments in cells where translation is inhibited, with in vitro refolded RNA, and in silico analyses of the thermodynamics of folding. Finally, they argue forcefully that for endogenous *E. coli* genes, mRNA structure is a more reliable predictor of translational efficiency than codon adaptation (tAI) or the recently published codon influence metric. The experiments are well controlled, clearly explained, and compelling, and their findings have important implications for gene expression in bacteria.

*Reviewer #2:*

This manuscript reports experiments that compare in vivo mRNA structure to translation efficiency in *E. coli*. The authors find a negative correlation between these measures for 1,100 genes. They then make similar comparisons to mRNA structure in vivo after treating with a drug that blocks translation initiation (700 genes), and in vitro denatured / refolded mRNA (400 genes), and find somewhat lower correlations. They found smaller correlations between translation efficiency and tAI, Shine-Dalgarno strength, and measures of codon usage from Boel et al. They also provide compelling evidence that reporter overexpression induces expression of corresponding amino-acid synthesis pathways. The topic is important and timely, the paper is very well written, and the results are interesting. However, correlation is not causation, and the authors don't provide statistical comparisons of the correlations. This and several other issues decrease my enthusiasm.

1) The crux of the authors argument is that rho values are higher for in vivo mRNA measures than they are for tAI, Boel codon values, etc. In the first paragraph of the subsection “Translation efficiency is less correlated with other mRNA features” they say these are "significantly" different. The authors need to provide statistical tests that show the differences are significant if they want to make this argument. I'm not sure if that's possible with Spearman's rho. Pearson's R values allow comparisons via Fisher's z-test, but may not be appropriate because TE isn't normally distributed. Also, they used different genes for each correlation (see #2), which might affect their results and complicates comparisons. The results would be stronger if they could compare these correlations more directly (same genes, statistical significance in differences).

2) The authors say their mRNA structure analysis is "genome-wide" (Abstract), "global" (subsection “Development of global RNA secondary structure determination in *E. coli*”, first paragraph), and covers "all" genes (subsection “Translation efficiency is highly correlated with ORF mRNA structure”, second paragraph, e.g.). This isn't accurate, as they only use 13% to 30% of the genes because they have a 15 read / nt threshold on DMS-seq. A careful reader will spot this in the figure legend; a casual reader will miss this. This should be more explicit in the text.

3) Using all genes with TE > 0.01 in their supplemental table (3,358 genes) gives correlations of rho = -0.42 (Gini_WT vs TE), 0.26 (tAI vs TE), 0.31 (Boel-multiple vs. TE) and 0.36 (Boel-ordinal vs. TE). The point is that when one does a genome-wide analysis using the authors data, the correlations are much closer.

4) The supplemental table makes it difficult to reproduce the authors results, as they don't show which genes were picked for each correlation test. This should be simple to address, by including the reads/nt for each experiment and / or using multiple sheets in the excel file.

5) Overall, the study would be more compelling if the authors developed a multiple regression model to explain TE using mRNA structure, codon bias, SD sequence, SD pause sites (from their 2012 Nature paper), structure around start codons, etc. This approach would allow comparisons between these features, at least in terms of what makes a better predictor, and would result in a useful model for the community.

*Reviewer #3:*

In this study, structural probing of mRNA (DMS-seq) is combined with ribosome profiling to address the question of how translation initiation rate is normally tuned in *E. coli*. The authors find compelling evidence that each ORF within a given polycistronic mRNA represents an independent structural module, and the degree of structure inversely correlates with translation efficiency. In fact, ORF-wide secondary structure is a much better predictor of TE than other parameters (SD, tAI). This work draws important distinctions between the parameters that influence the efficiency of translation of exogenous (overexpressed) genes and those that govern translation of endogenous genes. The paper will be an eye-opener for many in the field.

Critique:

The experiments of Figure 6 show that a mutation that causes formation of an intergenic RNA structure inhibits translation of the downstream gene, due to occlusion of its start codon. While this lends strong support to the ORF domain model, it also begs the question of whether start codon occlusion helps tune initiation normally (in WT cells). The metagene analysis indicates that nucleotides near the start codon tend to be unpaired. The authors should address whether pairing probability of this region (start codon and immediate vicinity) is related to TE (even if the answer is no).

In the title, change "influence" to "predict" or "reflect." As the authors point out in the Discussion, whether the ORF structure determines initiation rate (via increased standby sites, for example) or plays another role (protects unoccupied mRNA from endonucleases) remains unclear. The latter idea would more readily explain why the entire ORF exhibits structure.

---

## [Author Response]

*[…] Reviewer #2:*

*This manuscript reports experiments that compare in vivo mRNA structure to translation efficiency in E. coli. The authors find a negative correlation between these measures for 1,100 genes. They then make similar comparisons to mRNA structure in vivo after treating with a drug that blocks translation initiation (700 genes), and in vitro denatured / refolded mRNA (400 genes), and find somewhat lower correlations. They found smaller correlations between translation efficiency and tAI, Shine-Dalgarno strength, and measures of codon usage from Boel et al. They also provide compelling evidence that reporter overexpression induces expression of corresponding amino-acid synthesis pathways. The topic is important and timely, the paper is very well written, and the results are interesting. However, correlation is not causation, and the authors don't provide statistical comparisons of the correlations. This and several other issues decrease my enthusiasm.*

*1) The crux of the authors argument is that rho values are higher for in vivo mRNA measures than they are for tAI, Boel codon values, etc. In the first paragraph of the subsection “Translation efficiency is less correlated with other mRNA features” they say these are "significantly" different. The authors need to provide statistical tests that show the differences are significant if they want to make this argument. I'm not sure if that's possible with Spearman's rho. Pearson's R values allow comparisons via Fisher's z-test, but may not be appropriate because TE isn't normally distributed.*

Due to the non-normal distribution of TE, we chose to use bootstrapping to estimate the 95% confidence intervals for different Spearman’s ρ values. We used the K-S test to compare the bootstrapped distributions of two Spearman’s ρ values. We found that correlation between TE and Gini calculated in every condition is significantly stronger than correlation between TE and tAI or boel’s codon influence (all p-values < 2.2e-16). We have incorporated this analysis in the revised manuscript (Figure 4—figure supplement 2). Therefore, mRNA secondary structure level seems to be the strongest predictor of endogenous TE.

*Also, they used different genes for each correlation (see #2), which might affect their results and complicates comparisons. The results would be stronger if they could compare these correlations more directly (same genes, statistical significance in differences).*

We repeated our analyses on the 421 genes that passed our ≥15 reads/nt DMS-seq filter in all 3 datasets (wt cells in vivo, ksg-treated cells in vivo, and in vitro modified mRNA). The Spearman’s Rank Coefficients (ρ) for Gini indices, computationally predicted free folding energy (-ΔG/nt), tAI and codon influence (Boel et al., 2016) correlated to TE are very similar to those reported in Figure 3 and Figure 4 for the entire data set. In each case, the correlation coefficient with TE is higher for in vivo mRNA structure level (Gini index) than tAI and Boel’s codon influence index. These data are presented in Figure 4—figure supplement 2.

*2) The authors say their mRNA structure analysis is "genome-wide" (Abstract), "global" (subsection “Development of global RNA secondary structure determination in E. coli”, first paragraph), and covers "all" genes (subsection “Translation efficiency is highly correlated with ORF mRNA structure”, second paragraph, e.g.). This isn't accurate, as they only use 13% to 30% of the genes because they have a 15 read / nt threshold on DMS-seq. A careful reader will spot this in the figure legend; a casual reader will miss this. This should be more explicit in the text.*

We thank the reviewer for pointing this out. We realized that we had not explained why we applied an ≥15 reads per nt cutoff for DMS-seq. Our selection was based on comparing the reproducibility of replicate RNA structure determinations at different read densities. A plot of the median of Pearson’s R values when comparing WT DMS-seq replicates of ORFs with different read densities indicates that an average coverage depth of ≥ 15 reads/nt greatly improves data reproducibility (Figure 1). This coverage requirement was also reported in analyzing DMS-MaPseq data from yeast (Meghan et al., 2016). For *E. coli* genome-wide DMS-seq data, ~30% genes passed this filter due to the low abundance of some mRNAs, deficiency of DMS modification, and/or the experimental noise. The ksg-treated samples have the lowest sequence coverage and therefore lowest number of genes passing the filter, likely due to the degradation of mRNA when ribosomes run off after the treatment. We note that the analysis after kasugamycin treatment is possible only because of the high response strain (Δ*gcvB*) that we developed. We have added the explanation of the criteria for gene selection for DMS-seq data in the text. We indicate the number of ORFs in each analysis in the body of the figure and perform all analyses for both the ORFs passing the ≥ 15 DMS-seq reads/nt cutoff in each conditions and the set of 421 ORFs that passed the cutoff in all 3 conditions (DMS-seq of wt cells in vivo, ksg-treated cells in vivo, and in vitro modified mRNA).

*3) Using all genes with TE > 0.01 in their supplemental table (3,358 genes) gives correlations of rho = -0.42 (Gini_WT vs TE), 0.26 (tAI vs TE), 0.31 (Boel-multiple vs. TE) and 0.36 (Boel-ordinal vs. TE). The point is that when one does a genome-wide analysis using the authors data, the correlations are much closer.*

It is not at all surprising that the correlation of Gini index with TE is lower when the ≥15 reads/nt filter is dropped, as the structures determined at lower read density are noisy and poorly reproducible. Thus, applying this filter assures high quality data rather than biasing our sample. Interestingly, even when we use the Spearman’s ρ values calculated by the reviewer on the lower quality dataset, the correlation of in vivo Gini index against TE is still significantly stronger than that of tAI or Boel’s codon influence value against TE (for all comparisons, p-value < 2.2e-16 with statistical analysis described in 1).

*4) The supplemental table makes it difficult to reproduce the authors results, as they don't show which genes were picked for each correlation test. This should be simple to address, by including the reads/nt for each experiment and / or using multiple sheets in the excel file.*

Thank you for pointing this out. We are sorry for the confusion. We have updated the supplemental tables with the filter criteria and genes selected for specific comparison in the revised manuscript.

*5) Overall, the study would be more compelling if the authors developed a multiple regression model to explain TE using mRNA structure, codon bias, SD sequence, SD pause sites (from their 2012 Nature paper), structure around start codons, etc. This approach would allow comparisons between these features, at least in terms of what makes a better predictor, and would result in a useful model for the community.*

We have now developed such a model, which is presented in Figure 4—figure supplement 3. We show that inclusion of variables other than ORF-wide structure (Boel codon influence metric, tRNA adaptation index, and Shine-Dalgarno sequence strength) showed marginal improvement in the predictive power compared to the ORF-wide structure alone. Pearson correlation (R^2^) between predicted and measured TE increases from 0.55 in the structure-only model to 0.61 in the all-inclusive model.

*Reviewer #3: […] Critique:*

*The experiments of Figure 6 show that a mutation that causes formation of an intergenic RNA structure inhibits translation of the downstream gene, due to occlusion of its start codon. While this lends strong support to the ORF domain model, it also begs the question of whether start codon occlusion helps tune initiation normally (in WT cells). The metagene analysis indicates that nucleotides near the start codon tend to be unpaired. The authors should address whether pairing probability of this region (start codon and immediate vicinity) is related to TE (even if the answer is no).*

We plotted comparison between TE and the mRNA structural level, quantified by Gini index, of different regions in genes, including (1) from 20nt upstream of the first base of start codon to 40nt downstream (-20:40), (2) from the first base of start codon to 60nt downstream (0:60), (3) from 60nt downstream of start codon to the stop codon (60:end), and (4) full length open reading frame. For this analysis, we used the ORFs with ≥15 reads/nt of DMS-seq signal in all conditions analyzed (wt cells in vivo, ksg-treat cells in vivo and in vitro modified RNA), with the additional criterion that the ORF immediately upstream is at least 20nt away to prevent interference between overlapping genes (Figure 3—figure supplement 1).

The Gini index of start vicinity region is lower than that of the downstream region in all conditions analyzed, which agrees with our metagene analysis that the mRNA near the start codon tend to be more unpaired. The correlation between TE and Gini of start vicinity region (-20:40 and 0:60) is significantly lower than that between TE and full ORF Gini (p < 2.2e-16, K-S test comparing the two bootstrapped distributions of Spearman’s ρ). As we summarized in the Introduction, previous work showed that calculated mRNA structure around the start codon can affect gene translatability. However, our work shows that the overall ORF structure is much more predictive of translation efficiency than RNA structure at the start, even in ksg-treated cells where the effect of translating ribosome on mRNA structure is eliminated. Supporting this conclusion, the correlation between TE and Gini of gene body (60:end) is slightly lower than that between TE and full ORF Gini, but the difference is significant (p < 2.2e-16, K-S test comparing the two bootstrapped distributions of Spearman’s ρ).

*In the title, change "influence" to "predict" or "reflect." As the authors point out in the Discussion, whether the ORF structure determines initiation rate (via increased standby sites, for example) or plays another role (protects unoccupied mRNA from endonucleases) remains unclear. The latter idea would more readily explain why the entire ORF exhibits structure.*

Changed in the revised title.